# Longitudinal Metabolomic Analysis Reveals Gut Microbial-Derived Metabolites Related to Formula Feeding and Milk Sensitization Development in Infancy

**DOI:** 10.3390/metabo12020127

**Published:** 2022-01-28

**Authors:** Ching-Min Tang, Gigin Lin, Meng-Han Chiang, Kuo-Wei Yeh, Jing-Long Huang, Kuan-Wen Su, Ming-Han Tsai, Man-Chin Hua, Sui-Ling Liao, Shen-Hao Lai, Chih-Yung Chiu

**Affiliations:** 1Department of Pediatrics, Chang Gung Memorial Hospital at Linkou, Chang Gung University, Taoyuan 333, Taiwan; sugarwahaha@adm.cgmh.org.tw (C.-M.T.); kwyeh@adm.cgmh.org.tw (K.-W.Y.); kpm104@adm.cgmh.org.tw (S.-H.L.); 2Department of Medical Imaging and Intervention, Chang Gung Memorial Hospital at Linkou, Chang Gung University, Taoyuan 333, Taiwan; giginlin@adm.cgmh.org.tw; 3Clinical Metabolomics Core Laboratory, Chang Gung Memorial Hospital at Linkou, Taoyuan 333, Taiwan; neo0914@adm.cgmh.org.tw; 4Department of Pediatrics, New Taipei Municipal TuCheng Hospital, Chang Gung Memorial Hospital, Chang Gung University, Taoyuan 333, Taiwan; long@adm.cgmh.com.tw; 5Department of Pediatrics, Chang Gung Memorial Hospital at Keelung, Chang Gung University, Taoyuan 333, Taiwan; b87401102@cgmh.org.tw (K.-W.S.); a12270@adm.cgmh.org.tw (M.-H.T.); menchin@adm.cgmh.org.tw (M.-C.H.); suiliao@adm.cgmh.org.tw (S.-L.L.)

**Keywords:** breastfeeding, formula feeding, milk sensitization, metabolomics, urine

## Abstract

Early exposure to formula milk increases the likelihood of cow’s milk sensitization and food allergies in the later childhood. However, the underlying mechanisms are multifactorial and unclear. Fifty-five children from a follow-up birth cohort study were grouped into exclusive breastfeeding (EBF, *n* = 33) and formula feeding (EFF, *n* = 22) in the first six months of life. Urinary metabolites were longitudinally assessed and analyzed at 6 months, 1, and 2 years of age using ^1^H-nuclear magnetic resonance (NMR) spectroscopy. Integrated analysis of metabolic profiling associated with formula feeding and milk sensitization related to IgE reactions was also investigated. Twenty-two metabolites were significantly obtained in the EFF set at age 0.5, whereas nine metabolites were predominantly obtained in the milk sensitization set at age 1. A subsequent analysis of metabolic change from 6 months to age 1 identified eight metabolites, including 3-methyl-2-oxovaleric acid, glutarate, lysine, N-phenylacetylglycine, N,N-dimethylglycine, 3-indoxysulfate, 2-oxoglutaric acid, and pantothenate associated with formula feeding and milk sensitization with same trend variation. Among them, 3-indoxysulfate, N-phenylacetylglycine, and N,N-dimethylglycine were gut microbial-derived without IgE association. By contrast, 3-methyl-2-oxovaleric acid, glutarate, and lysine were IgE related associated with formula feeding contributing to milk sensitization (*p* < 0.05). Longitudinal urinary metabolomic analysis provides molecular insight into the mechanism of formula feeding associated with milk sensitization. Gut microbial-derived metabolites associated with formula feeding and IgE associated metabolites related to branched-chain amino acid metabolism play roles in developing sensitization and allergic symptoms in response to formula feeding.

## 1. Introduction

Human milk is considered the best nutritional source for infants, and it contains numerous biologically active agents that benefit in many ways [1]. Particularly, breast milk helps in shaping the gut microbiota with a stable and healthy early gut microbial ecosystem for later allergy prevention [2]. However, breastfeeding may not be always available, and formula milk is an effective substitution for breast milk to satisfy the infant’s nutrition and growth [3].

The formula milk is industrially designed to mimic the nutritional profile of breast milk. However, the high ratio of casein to whey proteins makes it harder to digest, and its casein protein fractions have been linked to food allergens and allergies during early life [4,5]. In previous studies, children with exclusive breastfeeding are associated with reduced cow’s milk sensitization and eczema in early childhood [6,7]. Avoiding infants’ exposure to cow’s milk at birth appears to prevent cow’s milk sensitization and clinical food allergies [8]. Despite this association, the molecular mechanisms underlying the development of food sensitization by formula supplementation remain undetermined.

Metabolomics provides a comprehensive study of a wide variety of metabolites and their interactions at a molecular and cellular level. Using nuclear magnetic resonance (NMR) spectroscopy, it enables detailed characterization of metabolic rearrangements in an immune process and helps discover biomarkers involved in allergic response and allergies [9]. Clinically, urinary metabolomics primarily reflects food allergic reactions against allergies [10]. The major aim of the study was to identify the longitudinal metabolic profiles in urine of children fed with breast milk or with cow’s milk formulas by using ^1^H-NMR spectroscopy. A comprehensive metabolomics-based approach to address the impact of different breastfeeding patterns on milk sensitization is hypothesized to reveal insights into the molecular mechanisms of formula feeding affecting allergic reactions and provide a potential strategy for prevention and treatment of childhood allergies.

## 2. Results

### 2.1. Population Characteristics

A total of 165 urine samples collected from 33 exclusively breast-fed children and 22 formula-fed children were analyzed. Baseline characteristics of children in relation to exclusive breastfeeding and formula feeding are shown in (Table 1).

A significantly higher prevalence of milk sensitization was observed in children who were formula-fed compared to children who were breast-fed at the age of 1 year (*p* < 0.05). There was no difference in family atopy history, household income, and infant sex, maternal and gestational age, and birth body mass index (BMI).

### 2.2. Identification of Metabolites in Different Breastfeeding Pattern and Milk Sensitization Sets at Different Years of Age

^1^H NMR data of urinary metabolites obtained at 0.5, 1, and 2 years of age were collected and analyzed. Unsupervised principal components analysis (PCA) failed to separate groups clearly based on parental atopy, parental smoking, household income, and the sex of child (Appendix A). Appendix A show the two-dimensional graphs of PLS-DA score plots from the different datasets. A clear separation was found in the metabolites between exclusive breastfeeding and formula feeding (Q^2^/R^2^ > 0.5) at age 0.5 (Appendix A). Metabolites identified with an FDR-adjusted *p*-value < 0.05 in the change of expression level in different breastfeeding pattern and milk sensitization sets are shown in (Appendix A), respectively. A total of 22 metabolites were predominantly obtained at age 0.5 in the set of different breastfeeding pattern, whereas nine metabolites were predominantly obtained at age 1 in the milk sensitization set. Most importantly, seven of these nine metabolites were found to be significantly inversely expressed in children with formula feeding at age 0.5. Metabolic changes from 6 months to 1 year of age were therefore subsequently assessed.

### 2.3. Metabolites in Different Breastfeeding Pattern and Milk Sensitization Sets from 6 Months to 1 Year of Age

A good separation of PLS-DA score plots in different breastfeeding pattern and milk sensitization from 6 months to 1 year of age (1 y/6 m) is respectively shown in Appendix A. Table 2 shows the metabolites significantly differentially expressed in different breastfeeding patterns and milk sensitization sets from 6 months to 1 year of age. Eight common metabolites including 3-methyl-2-oxovaleric acid, glutarate, lysine, N-phenylacetylglycine, N,N-dimethylglycine, 3-indoxysulfate, 2-oxoglutaric acid, and pantothenate were significantly associated with formula feeding and milk sensitization with same trend variation. Among them, levels of N,N-dimethylglycine and 2-oxoglutaric acid were increased from 6 months to 1 year old, whereas the other six metabolites were reduced.

### 2.4. Association between Metabolites and Food Allergen-Specific IgE Levels in Different Breastfeeding Patterns and Milk Sensitization Sets

Correlations between food allergen-specific IgE levels and metabolites significantly expressed in different breastfeeding patterns and milk sensitization sets at age 0.5 and age 1 are shown in Figure 1a,b, respectively. At age 0.5, seven metabolites were significantly and positively correlated with milk-specific IgE levels, whereas four metabolites were significantly correlated with egg-white IgE levels (Figure 1a). In contrast, there was no significant correlation between metabolites associated with milk sensitization and food allergen-specific IgE levels at age 1 (Figure 1b). However, from 6 months to 1 year of age, the changes of levels of five metabolites including 3-methyl-2-oxovaleric acid, lysine, allantoin, glutarate and dimethyl sulfone were significantly positively correlated with the changes of total serum IgE levels (Figure 1c).

### 2.5. Correlations between Metabolites of Formula Feeding and of Milk Sensitization

Spearman correlation coefficient analyses were performed to determine the correlations between metabolites of different breastfeeding patterns and milk sensitization sets from age 0.5 to age 1. Six of the eight metabolites simultaneously associated with formula feeding and the milk sensitization were strongly correlated with metabolites only associated with formula feeding (Figure 2). Among them, three metabolites including 3-methyl-2-oxovaleric acid, lysine, and glutarate, which significantly positively associated with IgE, were strongly correlated with each other (*p* < 0.01).

### 2.6. Metabolic Pathway and Functional Analysis

Metabolic pathways of metabolites both significantly associated with formula feeding and milk sensitization were analyzed using MetaboAnalyst webserver (Table 3). Metabolites between non-IgE-related (2-oxoglutaric acid) and IgE-related (lysine) associated with milk sensitization were significantly associated with amino acid metabolisms. Cofactors and vitamin metabolisms including pantothenate and CoA biosynthesis and biotin metabolism were significantly associated with non-IgE related and IgE-related metabolites related to milk sensitization, respectively. Figure 3 shows the metabolic pathways of metabolites significantly associated with formula feeding related to milk sensitization.

## 3. Discussion

Infant formula is designed as an effective substitution to mimic the nutritional profile of breast milk for infants that cannot be breastfed. However, early exposure to formula milk was reported with an increased probability of cow’s milk sensitization and future food allergies. The underlying molecular mechanisms are still undetermined. Using metabolomic technology, this study has demonstrated a strong correlation between the altered gut microbiota composition after formula feeding and the further milk sensitization. Furthermore, TCA cycle activation and IgE-related branched-chain amino acids metabolism were also associated with the milk sensitization development.

Indole derivatives could be produced from the essential amino acid tryptophan by gut microbiota [11], and help maintain intestinal barrier integrity and immune cell homeostasis [12]. Clinically, breastfeeding shapes the gut bacterial community with long-term health implications [13], while infant formula could change gut bacteria in early life and contribute to childhood allergy [14,15]. In this study, 3-indoxysulfate was found to be significantly lower in formula-fed infants with milk sensitization at age 1. These results suggest that the gut microbiota and dysbiosis related to formula may take part in the reduction of indole production and a reduction in intestinal barrier integrity for allergen sensitization.

Phenylacetylglycine was known as a gut microbial metabolite converted from phenylalanine in the cecum [16]. The literature review represents that infants fed with formula milk have higher plasma concentration of phenylalanine than those fed with breast milk [3,17]. Phenylacetylglycine was reported to be correlated with mucosal antibody responses, linking the microbial metabolites to immunity influence [18]. In this study, phenylacetylglycine level elevated in the formula-fed infants but longitudinally decreased in infants with milk sensitization. This result implies that formula feeding may alter gut microbiota composition and affect the microbial-host metabolic profile contributing to the following immune response to allergen exposure.

Dimethylglycine (DMG), a tertiary amino acid derived from choline, has been known to be beneficial in many ways to the cell [19]. As in previous reports [20], urinary DMG was significantly decreased in formula-fed infants in this study. However, a decreased DMG level was strongly associated with the prevalence of milk sensitization but not the IgE levels. DMG provides a protective role in airway inflammation via inhibiting the release of type-2 T-helper cytokines and leukotriene [21]. Our findings indicate that infants with formula feeding may decrease urinary DMG, contributing to active or increase Th2 cell immunity and subsequent allergic sensitization and allergy.

3-Methyl-2-oxovaleric acid participates in the metabolism of branched amino acid [22], generated from isoleucine and glycine, and correlates to the lysine metabolism and tricarboxylic acid [23] cycle through oxaloacetate [24,25]. In this study, 3-methyl-2-oxovaleric acid, lysine and glutarate were all predominately measured in the urine of formula-fed infants and observed a significant decrease in infants with milk sensitization correlated with total serum IgE levels. This finding supports that branched-chain amino acid metabolism impacts allergic response to food and allergies [10]. Lysine is an important amino acid for IgE binding in the epitope of allergens [26], which may particularly explain the significant consumption of lysine in formula-fed infants associated with IgE-related lysine metabolism in this study.

2-Oxoglutaric acid and pantothenate participating in the pathway of TCA cycle [27] were significantly associated with formula-fed infants related to milk sensitization in this study. The accumulation of intermediates of TCA cycle, such as citrate, succinate and fumarate, can act as signaling molecules to command cells that an inflammatory process has initiated [28]. An effective consumption of pantothenate and an accumulation of 2-oxoglutaric acid in infants with formula feeding may regulate the TCA cycle by activating inflammatory responses, which contributes to the development of allergen sensitization.

The major limitation of the study is its relatively small sample size with limited statistical power for subanalyses. However, the major strength of this study lies in its longitudinal metabolomic analysis, which allows dynamic assessments for metabolic changes in the development of milk sensitization related to different breastfeeding patterns.

In conclusion, urinary metabolomic analysis reveals pathways participated in the process of formula feeding contributing to milk sensitization. Metabolites related to branched-chain amino acid metabolism and cofactor and vitamin metabolism for TCA cycle activation appears to be associated with allergic IgE production and the inflammatory process against milk sensitization. Furthermore, the identification of gut microbial-derived metabolites associate with milk sensitization implies the effect of formula feeding on gut microbiota composition and mucosal immune responses in developing allergen sensitization. The implication of these findings is that formula additives with rich branched-chain amino acid and probiotics and/or prebiotics may prevent milk sensitization or subsequent allergies in formula-fed children. However, an integrative analysis of the intestinal microbiome and metabolome with a larger sample size is required to examine the functional significance of these observations providing comprehensive information to uncover mechanisms of microbial-associated pathogenic changes for milk sensitization in formula-fed children.

## 4. Materials and Methods

### 4.1. Study Population

A longitudinal study was performed to investigate the urinary metabolic profile of children from a birth cohort in the Prediction of Allergies in Taiwanese Children (PATCH) study. The subject recruitment in this birth cohort was reported previously, with 182 of 258 (70.5%) children regularly followed up at the clinic. Subjects dropped out during the follow-up period were excluded. Breastfeeding or specialized infant formula is recommended to be continued into the second year of life [29]. For longitudinal analysis, metabolic profile data of children since birth to 2 years of age were therefore collected and analyzed. Demographic data including personal history of enrolled children, breastfeeding condition, family atopic history, smoking exposure, and household income were collected. This study was approved by the Ethics Committee of Chang Gung Memory Hospital (No. 102-1842C). Written Informed consent was obtained from the parents or guardians of all study subjects.

### 4.2. Definition of Breastfeeding History

The detailed information regarding the pattern of breastfeeding was obtained by well-trained investigators at 6 months of age. As described previously [7], infants who were fed breast milk only without additional foods or drinks except water were defined as exclusively breast-fed (EBF). In contrast, infants who were fed formula only were exclusively formula-fed (EFF). Breastfeeding status was classified into two groups: EBF for 6 months and EFF for 6 months. Of the 182 children, seventy-seven children who were mixed fed in the first 6 months of life were excluded. Thirty-three exclusively breast-fed children and twenty-two formula-fed children having urine samples taken over at all time-points during infancy were enrolled and analyzed.

### 4.3. Total Serum and Food Allergen-Specific IgE Level Measurement

Serum level of total immunoglobulin E and allergen-specific IgE were measured by ImmunoCAP (Phadia, Uppsala, Sweden) and a commercial assay for IgE (ImmunoCAP Phadiatop Infant; Phadia) respectively at 6 months, and 1 and 2 years of age. Two most common food allergens, cow’s milk, and egg white were measured. Allergic sensitization was defined as ImmunoCAP Phadiatop Infant values ≥ 0.35 kU/L [30].

### 4.4. Urine Sample Preparation

Spot urine samples collected in the morning at age 6 months, 1 and 2 years were prepared and processed for spectrum acquisition as previously described [31]. Briefly, after thawing to 4 °C, a 900 μL pool of urine was treated with 100 μL of 1.5 M phosphate buffer (pH 7.4) in deuterium water containing 0.04% TSP [3-(trimethylsilyl)-propionic-2,2,3,3-d4 acid sodium salt] as an internal chemical shift reference standard. Each sample was vortexed for 20 s following centrifugation at 12,000× *g* for 30 min at 4 °C. Eventually, a total of 650 μL aliquot of the remaining supernatant was transferred to a standard 5-mm NMR tube for further experiments.

### 4.5. ^1^H-Nuclear Magnetic Resonance (NMR) Spectroscopy

NMR spectra were acquired at 300 k on a Bruker Avance 600 MHz spectrometer (Bruker BioSpin GmbH, Karlsruhe, Germany) equipped with a 5-mm CPTCI ^1^H cryoprobe located at Chang Gung Healthy Aging Research Center, Taiwan. A total of 64 scans were collected into 64 K computer data points for NMR spectra with a spectral width of 10,000 Hz (10 ppm) during the relaxation time of 4 s. One-dimensional (1D) NMR spectra were applied prior to zero-filled Fourier transformation at an exponential line broadening of 0.3 Hz. Then, the resulting ^1^H-NMR spectra were manually phased, baseline corrected, and further referenced to TSP at 0.0 ppm for NMR chemical shifts using TopSpin 3.2 software (Bruker BioSpin, Rheinstetten, Germany).

### 4.6. NMR Data Processing and Analysis

All raw ^1^H-NMR spectra data were processed and analyzed as previously mentioned [32]. The regions of residual water (δ4.745–4.845 ppm) and urea resonances (δ 5.465–6.195 ppm) were excluded to avoid spectral interference. Spectra normalization was rescaled to the creatinine integral at δ3.045 ppm to compensate for variations in urine concentrations [33]. All metabolites were identified using Chenomx NMR Suite 8.1 software (Chenomx, Edmonton, AB, Canada) with reference spectra from the Human Metabolome Database. The normalized ^1^H-NMR bucket data were transformed using generalized log transformation (glog) and uploaded to MetaboAnalyst 5.0 (http://www.metaboanalyst.ca (accessed on 18 October 2021)) for partial least squares-discriminant analysis (PLS-DA) to identify the metabolites used for discrimination between the groups. The Kyoto Encyclopedia of Genes and Genomes database (KEGG) were then employed to analyze their metabolic functional pathways.

### 4.7. Statistical Analysis

The baseline characteristics between children with different breastfeeding patterns were compared with univariate parametric and nonparametric tests. Differences in metabolites between two groups were analyzed with Mann-Whitney test with the MetaboAnalyst web server. A false discovery rate (FDR) of 5% was applied to correct for multiple tests. The correlation coefficients between differentially expressed metabolites and total serum and allergen-specific IgE levels were assessed using Spearman’s correlation test in R software (Lucent Technologies, NJ, USA, version 3.3.1). Statistical analysis was performed using IBM SPSS Statistics v. 20 [10]. All tests were two-tailed, and a *p* value < 0.05 was considered significant.

## Figures and Tables

**Figure 1 metabolites-12-00127-f001:**
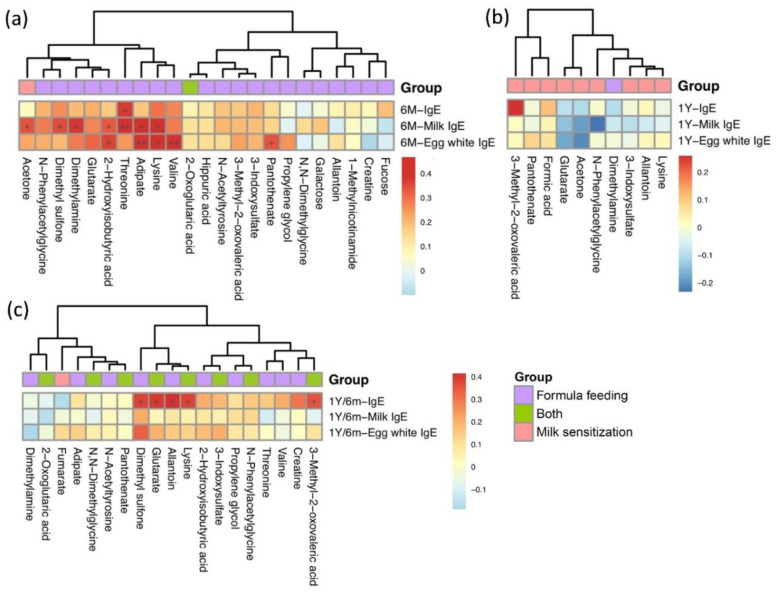
Heatmap of correlations between metabolites significantly differentially expressed in formula feeding and milk sensitization and total serum and food specific IgE levels at age 6 months (**a**), age 1 (**b**), and from 6 months to 1 year of age (**c**). Color intensity represents the magnitude of correlation. Red color represents positive correlations; blue color represents negative correlations. + symbol means a *p*-value < 0.05; ++ symbol means a *p*-value < 0.01.

**Figure 2 metabolites-12-00127-f002:**
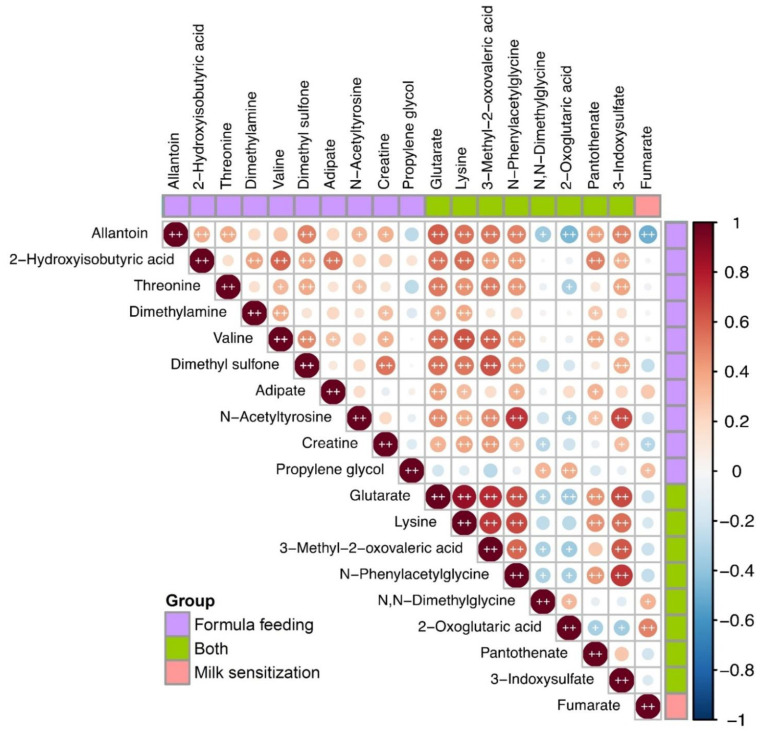
Heatmap of Spearman’s rank correlation coefficients of metabolites significantly differentially expressed in formula feeding and milk sensitization from 6 months to age 1. Color intensity represents the magnitude of correlation. Red color represents positive correlations; blue color represents negative correlations. + symbol means a *p*-value < 0.05; ++ symbol means a *p*-value < 0.01.

**Figure 3 metabolites-12-00127-f003:**
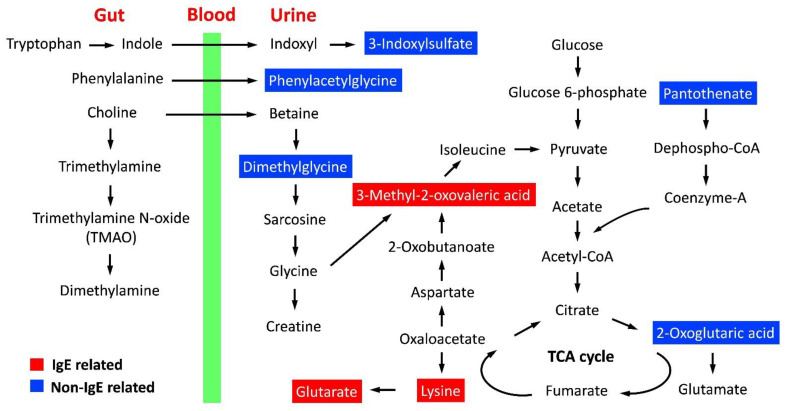
Schematic overview of metabolic pathways of eight metabolites both associated with formula feeding and milk sensitization. Identified metabolites significantly associated with IgE are shown in red color. TCA, tricarboxylic acid.

**Table 1 metabolites-12-00127-t001:** Baseline characteristics of 55 children in relation to exclusive breastfeeding and formula feeding.

Characteristics	Breastfeeding (*n* = 33)	Formula Feeding (*n* = 22)	*p* Value
**Family**			
Maternal atopy	16 (48.5%)	9 (40.9%)	0.580
Paternal atopy	18 (54.5%)	10 (45.5%)	0.509
Parental smoking	11 (33.3%)	12 (54.5%)	0.118
Household income			
Low, <500,000 NTD	13 (39.4%)	12 (54.5%)	0.517
Medium, 500,000–1,000,000 NTD	15 (45.5%)	8 (36.4%)	
High, >1,000,000 NTD	5 (15.2%)	2 (9.1%)	
**Infant**			
Sex, male (%)	19 (57.6%)	15 (68.2%)	0.428
Maternal age (yr)	31.2 ± 4.3	30.3 ± 4.9	0.442
Gestational age (wk)	38.5 ± 1.6	38.2 ± 1.8	0.578
Birth BMI (kg/m^2^)	12.1 ± 1.1	13.6 ± 4.3	0.123
Milk sensitization			
6 mo	4 (15.4%)	3 (23.1%)	0.666
1 yr	7 (26.9%)	11 (57.9%)	**0.036**
2 yr	11 (33.3%)	13 (59.1%)	0.106

Data shown are mean ± s.d. or number (%) of patients as appropriate. NTD, new Taiwan dollar; yr, year; wk, week; BMI, body mass index; mo, month. All *p* values < 0.05, which is in bold, are significant.

**Table 2 metabolites-12-00127-t002:** The VIP score and fold change of metabolites significantly differentially expressed categorizing by breastfeeding patterns and milk sensitization from 6 months to 1 year of age.

		Formula Feeding		*p*	Milk Sensitization		*p*
Metabolites	Chemical Shift, ppm (Multiplicity ^a^)	VIP Score ^b^	Fold change ^c^		VIP Score	Fold Change	
Allantoin	5.39–5.40 (s)	2.22	0.48	**0.001**	1.77	0.75	0.060
2-Hydroxyisobutyric acid	1.35–1.37 (s)	1.01	0.83	**0.004**	0.22	0.99	0.658
Threonine	4.25–4.27 (d)	1.16	0.79	**0.004**	0.58	0.88	0.327
Dimethylamine	2.71–2.73 (s)	0.92	0.85	**0.006**	0.48	1.07	0.321
Valine	1.04–1.05 (d)	0.87	0.85	**0.014**	0.42	0.96	0.406
Dimethyl sulfone	3.15–3.16 (s)	1.13	0.77	**0.015**	0.24	0.93	0.714
Adipate	1.55–1.56 (m)	1.15	0.72	**0.018**	0.39	1.19	0.576
N-Acetyltyrosine	7.16–7.18 (d)	1.63	0.51	**0.020**	1.56	0.59	0.114
Creatine	3.93–3.94 (s)	1.34	0.75	**0.043**	0.92	0.98	0.325
Propylene glycol	1.14–1.15 (d)	1.47	1.29	**0.049**	1.76	1.51	0.091
Glutarate	1.76–1.80 (tt)	2.11	0.63	**<0.001**	1.26	0.81	**0.039**
Lysine	1.89–1.91 (m)	1.75	0.70	**<0.001**	1.14	0.83	**0.039**
3-Methyl-2-oxovaleric acid	1.10–1.11 (d)	2.07	0.58	**<0.001**	1.81	0.68	**0.009**
N-Phenylacetylglycine	7.41–7.45 (s)	2.54	0.38	**<0.001**	2.59	0.43	**0.005**
N,N-Dimethylglycine	2.93–2.93 (s)	1.66	1.52	**0.001**	1.94	1.44	**0.004**
3-Indoxysulfate	7.50–7.52 (d)	2.06	0.45	**0.003**	2.25	0.48	**0.024**
2-Oxoglutaric acid	3.00–3.01 (t)	1.32	1.45	**0.008**	1.95	1.56	**0.004**
Pantothenate	0.93–0.94 (d)	1.04	0.81	**0.010**	1.17	0.85	**0.039**
Fumarate	6.52–6.53 (s)	1.58	1.41	0.054	2.41	1.55	**0.033**

^a^ Multiplicity, s, singlet; d, doublet; m, multiplet; t, triplet; tt, triplet of triplets. One of the proton assignments of the metabolite without signal overlapping was selected and the integral of selected peak was presented as a range of chemical shift. ^b^ VIP score were obtained from PLS-DA. ^c^ Fold change was calculated by dividing the value of metabolites in children with formula feeding by breastfeeding, and with by without milk sensitization from 6 months to 1 year of age (1 y/6 m). All FDR-adjusted *p* values < 0.05, which is in bold, are significant. VIP, Variable Importance in Projection.

**Table 3 metabolites-12-00127-t003:** Metabolic pathway and function analysis of metabolites associated with formula feeding related to milk sensitization.

Pathway	Metabolites	Pathway Name	Total	Hits	Raw *p*	FDR	Function
Non-IgE related	2-Oxoglutaric acid	D-glutamine and D-glutamate metabolism	6	1	0.015	0.85	Amino acid metabolism
		Arginine biosynthesis	14	1	0.036	0.85	Amino acid metabolism
		Butanoate metabolism	15	1	0.038	0.85	Carbohydrate metabolism
	Pantothenate	Pantothenate and CoA biosynthesis	19	1	0.048	0.85	Metabolism of cofactors and vitamins
IgE-related	Lysine	Biotin metabolism	10	1	0.019	1.00	Metabolism of cofactors and vitamins
		Lysine degradation	25	1	0.048	1.00	Amino acid metabolism

Total is the total number of compounds in the pathway; the Hits is the actually matched number from the user uploaded data; the Raw *p* is the original *p* value calculated from the enrichment analysis; the FDR is the portion of false positives above the user-specified score threshold. FDR, false discovery rate.

## Data Availability

Data is contained within the article.

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
