# Peer review of "Longitudinal Metabolomic Analysis Reveals Gut Microbial-Derived Metabolites Related to Formula Feeding and Milk Sensitization Development in Infancy"

_metabolites, 2022, doi:10.3390/metabo12020127_

Round 1
Reviewer 1 Report
There are still two of my comments which are not adequately explained in the revised version of the manuscript.
1. Table 2
“The assignment of the protons of each metabolite should be denoted. Chemical shifts cannot be presented in the third decimal. Furthermore, the exact chemical shift should be presented even in the case of multiple resonances and not as a range of chemical shifts.”
The authors could read any classical NMR book; they could not find even a single table with presentation of the chemical shifts to the third decimal, or as a range of chemical shifts, or without definition of the assignment. Furthermore, accuracy in the chemical shift to the third decimal and the range of chemical shifts up to 0.3 ppm are mutually inconsistent.
2. Page 10, lines 287-289
“Spectra normalization was rescaled to the creatinine integral at δ3.045 ppm to compensate for variations in urine concentrations.”
The authors did not adequately explain why the creatinine integral was used as a reference and not simply to add the well-established reference compound TSP-d4 which can provide accurate chemical shifts and integrals.
Based on (1) and (2), major revision is requested.
Author Response
Submission ID: metabolites-1549505
Title: Longitudinal Metabolomic Analysis Reveals Gut Microbial-Derived Metabolites Related to Formula Feeding and Milk Sensitization Development in Infancy
Metabolites
Reviewer 1:
There are still two of my comments which are not adequately explained in the revised version of the manuscript.
- Table 2, “The assignment of the protons of each metabolite should be denoted. Chemical shifts cannot be presented in the third decimal. Furthermore, the exact chemical shift should be presented even in the case of multiple resonances and not as a range of chemical shifts.” The authors could read any classical NMR book; they could not find even a single table with presentation of the chemical shifts to the third decimal, or as a range of chemical shifts, or without definition of the assignment. Furthermore, accuracy in the chemical shift to the third decimal and the range of chemical shifts up to 0.3 ppm are mutually inconsistent.
Ans: As reviewer said, the assignment of the protons of each metabolite should be denoted as in the classical NMR book. For example, as shown in the Figure below, Valine, which is identified by Chenomx Inc. software, can be denoted at 3.6, 2.3, 1.1, and 1.0 ppm.
However, most of these sites of ppm could be overlapped by protein or other metabolites. For clinical medicine, one site of denoted ppm without other metabolites overlapping will be chosen to compare between groups. As shown in the Figure below, valine denoted at 1.0 around (few chemical shit in each samples) will be chosen as the site of chemical shift of valine for comparison.
NMR spectrum of each sample will be aligned and the integral of each metabolite will be calculated. Also, as in the Figure below, each metabolite will have different peaks (multiplicity) with a range of chemical shift to present the integral of metabolite, such as valine 1.04-1.05(d) in this study.
Also, as mentioned in the method, a total of 64 scans were collected into 64 K computer data points for NMR spectra with a spectral width of 10,000 Hz (10 ppm), for this reason, the chemical shifts used for denotation of each metabolite will be presented in the second decimal (scales for bucking) with a range of chemical shifts for the integral of metabolite. The value used third decimal chemical shit is the mean of the range for single peak. However, to be more precise, all the presentation of chemical shift in Table 2 has amended to the range of the chemical shift as appropriate.
- Page 10, lines 287-289
“Spectra normalization was rescaled to the creatinine integral at δ3.045 ppm to compensate for variations in urine concentrations.” The authors did not adequately explain why the creatinine integral was used as a reference and not simply to add the well-established reference compound TSP-d4 which can provide accurate chemical shifts and integrals.
Ans: As reviewer said, TSP-d4 is well-established to be a reference to normalize the NMR spectrum data for analysis. However, for clinical medicine research especially by using urine sample, it is important to adjust the urine concentrations due to individual’s hydration status. Generally, urinary creatinine is often used to adjust for urine analyte concentrations because of its relative stability within an individual. Most importantly, each metabolite including creatinine will not be influenced by adjusting TSP-d4 or not if theses metabolites are adjusted by creatinine. For example, metabolite/TSP-d4 divided by creatinine/TSP-d4 = metabolite/creatinine.
Thanks for your precious comments and suggestions. Hope these responses could answer all your questions. Thanks again !!!
Sincerely
Chih-Yung Chiu MD. PhD.
Department of Pediatrics, Chang Gung Memorial Hospital at Linkou
5, Fuxing St., Guishan Dist., Taoyuan, Taiwan
Tel: 886-3-3281200 ext 8202; E-mail: pedchestic@gmail.com

Reviewer 2 Report
-
Author Response
Submission ID: metabolites-1549505
Title: Longitudinal Metabolomic Analysis Reveals Gut Microbial-Derived Metabolites Related to Formula Feeding and Milk Sensitization Development in Infancy
Metabolites
Reviewer 2:
--
Ans: Deeply thanks for your previous precious comments and suggestions.
Sincerely
Chih-Yung Chiu MD. PhD.
Department of Pediatrics, Chang Gung Memorial Hospital at Linkou
5, Fuxing St., Guishan Dist., Taoyuan, Taiwan
Tel: 886-3-3281200 ext 8202; E-mail: pedchestic@gmail.com

Reviewer 3 Report
Thank you for the responses to my comments. There remain some points to clarify.
- It would be useful in Figure S1 to identify the EBF and EFF individuals who were milk sensitive or not. Indeed, throughout the data, I think it is important to separate these 4 groups.
- The revised manuscript has only a minor change to the summary and has not addressed the comment about expanding the discussion to be more mechanistic and to include the implications of the findings. Also, what might future studies involve?
- The authors have added a further aim, rather than a hypothesis. What did they expect to find in their study, based on information in the Introduction?
- Although there were no differences in the numbers of male and female infants in each group, there could have been an influence of infant sex and it was good that this was assessed in the PCA analysis. Please add that information to the revised manuscript.
Author Response
Submission ID: metabolites-1549505
Title: Longitudinal Metabolomic Analysis Reveals Gut Microbial-Derived Metabolites Related to Formula Feeding and Milk Sensitization Development in Infancy
Metabolites
Reviewer 3:
Thank you for the responses to my comments. There remain some points to clarify.
1.It would be useful in Figure S1 to identify the EBF and EFF individuals who were milk sensitive or not. Indeed, throughout the data, I think it is important to separate these 4 groups.
Ans: As reviewer said, the information obtained from EBF and EFF with/without milk sensitization may provide important molecular mechanism for this study. However, the major limitation of this study is its relatively small sample size with limited statistical power for subanalyses. For example, the milk-sensitized children with EBF and EFF was only identified in 4, and 3 children respectively at age 0.5; 7, and 11 children respectively at age 1; 7, and 10 children respectively at age 2. The results obtained from inadequate sample size of this study may not provide useful information. By contrast, further studies with a larger sample size are important and required to examine the functional significance of our findings more comprehensively. Information regarding this has amended in the limitation section of Discussion and shown below:
In the Limitation section of Discussion,
“The major limitation of the study is its relatively small sample size with limited statistical power for subanalyses. However, the major strength of this study lies in its longitudinal metabolomic analysis, which allows dynamic assessments for metabolic changes in the development of milk sensitization related to different breastfeeding patterns.”
In the Conclusion section of Discussion,
“…….The implication of these findings is that formula additives with rich branched-chain amino acid and probiotics and/or prebiotics may prevent milk sensitization or subsequent allergies in formula-fed children. However, an integrative analysis of the intestinal microbiome and metabolome with a larger sample size is required to examine the functional significance of these observations providing comprehensive information to uncover mechanisms of microbial-associated pathogenic changes for milk sensitization in formula-fed children.”
Furthermore, PLS-DA score plots categorized by EBF +/- milk sensitization and EFF +/- milk sensitization 4 groups were shown below. However, it was difficult for reader to recognize the difference between groups and children were predominantly classified by EBF and EFF groups as shown in our previous Supplementary Figures. We suggest that it will be better to keep our previous presentation of PLS-DA score plots from the analysis of urine 1H-NMR spectra among children in different breastfeeding patterns (a), with and without milk sensitization (b) at age 0.5, 1, and 2.
2.The revised manuscript has only a minor change to the summary and has not addressed the comment about expanding the discussion to be more mechanistic and to include the implications of the findings. Also, what might future studies involve?
Ans: In this study, for the discussion of the potential biology and mechanism of metabolites related to formula feeding and milk sensitization, we try not to be more cautious with the conclusions and make sure the discussion are always supported by the results. Despite this, as reviewer’s suggestion, the implications of our findings and future studies involved have added into the Discussion section as below:
“In conclusion, urinary metabolomic analysis reveals pathways participated in the process of formula feeding contributing to milk sensitization. Metabolites related to branched-chain amino acid metabolism and cofactor and vitamin metabolism for TCA cycle activation appears to be associated with allergic IgE production and the inflammatory process against milk sensitization. Furthermore, the identification of gut microbial-derived metabolites associate with milk sensitization implies the effect of formula feeding on gut microbiota composition and mucosal immune responses in developing allergen sensitization. The implication of these findings is that formula additives with rich branched-chain amino acid and probiotics and/or prebiotics may prevent milk sensitization or subsequent allergies in formula-fed children. However, an integrative analysis of the intestinal microbiome and metabolome with a larger sample size is required to examine the functional significance of these observations providing comprehensive information to uncover mechanisms of microbial-associated pathogenic changes for milk sensitization in formula-fed children.”
3.The authors have added a further aim, rather than a hypothesis. What did they expect to find in their study, based on information in the Introduction?
Ans: As reviewer’s suggestion, a hypothesis for this study has added and shown below:
In the Introduction section,
“…………..The major aim of the study was to identify the longitudinal metabolic profiles in urine of children fed with breast milk or with cow's milk formulas by using 1H-NMR spectroscopy. A comprehensive metabolomics‐based approach to address the impact of different breastfeeding patterns on milk sensitization is hypothesized to reveal insights into the molecular mechanisms of formula feeding affecting allergic reactions and provide a potential strategy for prevention and treatment of childhood allergies.”
4.Although there were no differences in the numbers of male and female infants in each group, there could have been an influence of infant sex and it was good that this was assessed in the PCA analysis. Please add that information to the revised manuscript.
Ans: As we mentioned before, further subgroup analysis of sex for PLS-DA parameter tests (R2 and Q2) showed no clear separation in the metabolites between male and female (Q2/R2 < 0.5). As reviewer’s suggestion, the factor of sex was assessed in the unsupervised principal components analysis (PCA) and failed to separate groups clearly based on the sex of child. Information regarding this has added in the Supporting Information and shown below:
In the Identification of Metabolites in Different Breastfeeding Pattern and Milk Sensitization Sets at Different Years of Age section of Results,
“1H NMR data of urinary metabolites obtained at 0.5, 1, and 2 years of age were collected and analyzed. Unsupervised principal components analysis (PCA) failed to separate groups clearly based on parental atopy, parental smoking, house income, and the sex of child (Figure S1). Figure S2a and b……..”
Thanks for your precious comments and suggestions. Hope these responses could answer all your questions. Thanks again !!!
Sincerely
Chih-Yung Chiu MD. PhD.
Department of Pediatrics, Chang Gung Memorial Hospital at Linkou
5, Fuxing St., Guishan Dist., Taoyuan, Taiwan
Tel: 886-3-3281200 ext 8202; E-mail: pedchestic@gmail.com

Round 2
Reviewer 1 Report
On Table 2 the authors present a range of chemical shifts and not the exact chemical shift with a specific 1H assignment. Table 2, therefore, should be revised. I should also emphasize that I will not accept any further revised version of the manuscript.
Author Response
Submission ID: metabolites-1549505.R1
Title: Longitudinal Metabolomic Analysis Reveals Gut Microbial-Derived Metabolites Related to Formula Feeding and Milk Sensitization Development in Infancy
Metabolites
Reviewer 1:
1.On Table 2 the authors present a range of chemical shifts and not the exact chemical shift with a specific 1H assignment. Table 2, therefore, should be revised. I should also emphasize that I will not accept any further revised version of the manuscript.
Ans: As reviewer concerned that a range of chemical shifts may not the exact chemical shift with a specific 1H assignment. As the NMR spectra obtained from HMDB, for example, lactic acid was denoted at 4.10, and at 1.32 ppm. The site of ppm was shown in “Cluster Midpoint” to the second decimal. For the integral of selected peak, a range of chemical shift could be a way for further presentation.
Most importantly, for data analysis, most of proton assignments may be overlapped by protein or other metabolites. For clinical medicine in this study, one of the proton assignments of the metabolite without signal overlapping was selected and the integral of selected peak was presented as a range of chemical shift. Information regarding this has added and emphasized in the footnotes of Tables as below:
In the footnotes of Table 2:
“aMultiplicity, s, singlet; d, doublet; m, multiplet; t, triplet; tt, triplet of triplets. One of the proton assignments of the metabolite without signal overlapping was selected and the integral of selected peak was presented as a range of chemical shift. bVIP score were obtained from PLS-DA. cFold change was calculated by dividing the value of metabolites in children with formula feeding by breastfeeding, and with by without milk sensitization from 6 months to 1 year of age (1y/6m). All FDR-adjusted p values < 0.05, which is in bold, are significant. VIP, Variable Importance in Projection.’
Thanks for your precious comments and suggestions. Hope these responses could answer all your questions. Thanks again !!!
Sincerely
Chih-Yung Chiu MD. PhD.
Department of Pediatrics, Chang Gung Memorial Hospital at Linkou
5, Fuxing St., Guishan Dist., Taoyuan, Taiwan
Tel: 886-3-3281200 ext 8202; E-mail: pedchestic@gmail.com

This manuscript is a resubmission of an earlier submission. The following is a list of the peer review reports and author responses from that submission.
Round 1
Reviewer 1 Report
The paper describes a 1H NMR longitudinal metabolomic analysis of gut microbial-derived metabolites related to formula feeding and milk sensitization development in infancy. I could recommend this paper for publication subject to major revisions as indicated below:
- Table 2
The assignment of the protons of each metabolite should be denoted. Chemical shifts cannot be presented in the third decimal. Furthermore, the exact chemical shift should be presented even in the case of multiple resonances and not as a range of chemical shifts.
- Page 9, “4.5. 1H-Nuclear magnetic resonance (NMR) spectroscopy”
The acquisition time and relaxation delay should be explicitly defined since pulse repetition rate may affect the accuracy of the integrals of the resonances. Which are the T1 relaxation times of the analytes of interest?
- Page 10, lines 287-289
“Spectra normalization was rescaled to the creatinine integral at δ3.045 ppm to compensate for variations in urine concentrations.”
The authors should explain why the creatinine integral was used as a reference. How do they know that it remains constant for different samples?
- References
References are not presented in a uniform way. Several of them are presented without volume, year, and pages or even without the title of the journal. This is an unaccepted way of presenting references.
Reviewer 2 Report
In this manuscript, the authors present a metabolomic analysis that reveals gut microbial-derived metabolites related to formula feeding and milk sensitization development in infancy
In my opinion, the data presented in the present manuscript raise questions. For example:
|
1. In Table 1 is reported that at age 6m the children with milk sensitization are: |
|
|||
|
|
Breastfeeding (n = 33) |
Formula feeding (n = 22) |
|
|
|
6 mo |
4 (15.4%) |
3 (23.1%) |
0.666 |
|
|
1 yr |
7 (26.9%) |
11 (57.9%) |
0.036 |
|
|
2 yr |
11 (33.3%) |
13 (59.1%) |
0.106 |
|
That means that at age 6m, only 7 children from both groups are with milk sensitization. However, in Figure S1 (b) at Age 0.5 (6m), in the PLSDA model seems that the children with milk sensitization are about 35 plus 16 without sensitization. Is there something I did not understand correctly?
2.
Line 83: A clear separation was found in the metabolites between exclusive breastfeeding and formula feeding (Q2/R2 > 0.5) at age 0.5 (Table S1).
In Table S1 data are not clearly presented, for example: Q2 =0.40 corresponds to the comparison between which groups?
3.
Figure S1 c. In these patterns 4 groups are compared each time. For the first scores plot there are 2 clouds for the groups: Age 1, Age 0.5, breastfeeding and formula feeding. How this pattern is interpreted?
Reviewer 3 Report
This study characterises the urinary metabolome in infants breast-fed or fed formula milk for the first 6 months of life.
1. It was unclear whether the milk sensitisation group included both EBF and EFF infants together or whether they were analysed separately.
2. The Discussion is rather brief and could be expanded to link the findings (especially with milk sensitisation) and include possible mechanisms, future work and implications.
3. Include a hypothesis in the Introduction.
4. The manuscript mentions a 4-year follow-up. Were data available from all 4 years?
5. Were there any differences in the data between male and female infants?
6. Do the data relate to measurements made in faeces or blood, either in this or other studies (to demonstrate the relationship with gut microbiota)?